# Early and Late Response and Glucocorticoid-Sparing Effect of Belimumab in Patients with Systemic Lupus Erythematosus with Joint and Skin Manifestations: Results from the Belimumab in Real Life Setting Study—Joint and Skin (BeRLiSS-JS)

**DOI:** 10.3390/jpm13040691

**Published:** 2023-04-20

**Authors:** Margherita Zen, Mariele Gatto, Roberto Depascale, Francesca Regola, Micaela Fredi, Laura Andreoli, Franco Franceschini, Maria Letizia Urban, Giacomo Emmi, Fulvia Ceccarelli, Fabrizio Conti, Alessandra Bortoluzzi, Marcello Govoni, Chiara Tani, Marta Mosca, Tania Ubiali, Maria Gerosa, Enrica P. Bozzolo, Valentina Canti, Paolo Cardinaletti, Armando Gabrielli, Giacomo Tanti, Elisa Gremese, Ginevra De Marchi, Salvatore De Vita, Serena Fasano, Francesco Ciccia, Giulia Pazzola, Carlo Salvarani, Simone Negrini, Andrea Di Matteo, Rossella De Angelis, Giovanni Orsolini, Maurizio Rossini, Paola Faggioli, Antonella Laria, Matteo Piga, Alberto Cauli, Salvatore Scarpato, Francesca Wanda Rossi, Amato De Paulis, Enrico Brunetta, Angela Ceribelli, Carlo Selmi, Marcella Prete, Vito Racanelli, Angelo Vacca, Elena Bartoloni, Roberto Gerli, Elisabetta Zanatta, Maddalena Larosa, Francesca Saccon, Andrea Doria, Luca Iaccarino

**Affiliations:** 1Rheumatology Unit, Department of Medicine-DIMED, University of Padova, Via Giustiniani, 2, 35128 Padova, Italy; margherita.zen@unipd.it (M.Z.); luca.iaccarino@unipd.it (L.I.); 2ASST Spedali Civili di Brescia, Department of Clinical and Experimental Sciences, Rheumatology and Clinical Immunology, 25123 Brescia, Italy; 3Department of Experimental and Clinical Medicine, University of Florence, 50139 Firenze, Italy; 4Dipartimento Universitario di Scienze Cliniche, Internistiche, Anestesiologiche e Cardiovascolari (SCIAC) ‘Sapienza’ University, 00185 Rome, Italy; 5Rheumatology Unit, Azienda Ospedaliero-Universitaria S. Anna—Ferrara, University of Ferrara, 44121 Ferrara, Italy; 6Rheumatology, University of Pisa, 56124 Pisa, Italy; 7Clinical Rheumatology Unit Milano, ASST Gaetano Pini, Department of Clinical Sciences and Community Health, Lombardia, 20129 Milan, Italy; 8Unit of Immunology, Rheumatology, Allergy and Rare Diseases, IRCCS Ospedale San Raffaele, 20100 Milan, Italy; 9Dipartimento di Scienze Cliniche e Molecolari, Università Politecnica delle Marche, 60121 Ancona, Italy; 10Division of Rheumatology, Università Cattolica del Sacro Cuore Sede di Roma, 00168 Rome, Italy; 11Division of Rheumatology, Fondazione Policlinico Universitario A. Gemelli-IRCCS, 00168 Rome, Italy; 12Rheumatology Unit, University of Udine, Medical Area, 33100 Udine, Italy; 13Deparment of Precision Medicine Napoli, Università degli Studi della Campania Luigi Vanvitelli, 81100 Caserta, Italy; 14Azienda USL-IRCCS di Reggio Emilia, 42123 Reggio Emilia, Italy; 15Rheumatology Unit, University of Modena and Reggio Emilia, 41125 Reggio Emilia, Italy; 16Internal Medicine Unit, Department of Internal Medicine, Università degli Studi di Genova, 16146 Genoa, Italy; 17Deparment of Clinical and Experimental Sciences, Università Politecnica delle Marche, Rheumatology Clinic, 60131 Ancona, Italy; 18Unit of Rheumatology, University of di Verona, 37134 Verona, Italy; 19ASST OVEST Milanese Presidio di Legnano, 20025 Legnano, Italy; 20ASST OVEST Milanese Presidio di Magenta, 20013 Magenta, Italy; 21Rheumatology Unit, AOU University Clinic, University of Cagliari, 09124 Cagliari, Italy; 22Rheumatology, Ospedale M. Scarlato, Scafati, 84018 Salerno, Italy; 23Dipartimento di Scienze Mediche, Traslazionali e Centro di Ricerca Immunologia Base e Clinica (CISI), University of Napoli Federico II, 80131 Napoli, Italy; 24IRCCS Humanitas Research Hospital, Milan, Italy, 20089 Milan, Italy; 25Department of Rheumatology and Clinical Immunology, IRCCS Humanitas Research Hospital, 20089 Milan, Italy; 26Department of Biomedical Sciences, Humanitas University, 20133 Milan, Italy; 27Unit of Internal Medicine, Department of Biomedical Sciences and Human Oncology, University of Bari, 70125 Bari, Italy; 28Rheumatology Unit, Department of Medicine, University of Perugia, 06121 Perugia, Italy

**Keywords:** systemic lupus erythematosus, belimumab, remission, low disease activity, Cutaneous LE Area and Severity Index (CLASI), disease activity score (DAS)-28

## Abstract

Aim. To assess the efficacy of belimumab in joint and skin manifestations in a nationwide cohort of patients with SLE. Methods. All patients with skin and joint involvement enrolled in the BeRLiSS cohort were considered. Belimumab (intravenous, 10 mg/kg) effectiveness in joint and skin manifestations was assessed by DAS28 and CLASI, respectively. Attainment and predictors of DAS28 remission (<2.6) and LDA (≥2.6, ≤3.2), CLASI = 0, 1, and improvement in DAS28 and CLASI indices ≥20%, ≥50%, and ≥70% were evaluated at 6, 12, 24, and 36 months. Results. DAS28 < 2.6 was achieved by 46%, 57%, and 71% of patients at 6, 12, and 24 months, respectively. CLASI = 0 was achieved by 36%, 48%, and 62% of patients at 6, 12, and 24 months, respectively. Belimumab showed a glucocorticoid-sparing effect, being glucocorticoid-free at 8.5%, 15.4%, 25.6%, and 31.6% of patients at 6, 12, 24, and 36 months, respectively. Patients achieving DAS-LDA and CLASI-50 at 6 months had a higher probability of remission at 12 months compared with those who did not (*p* = 0.034 and *p* = 0.028, respectively). Conclusions. Belimumab led to clinical improvement in a significant proportion of patients with joint or skin involvement in a real-life setting and was associated with a glucocorticoid-sparing effect. A significant proportion of patients with a partial response at 6 months achieved remission later on during follow-up.

## 1. Introduction

Over ten years of experience in clinical practice has confirmed the efficacy and safety of belimumab in the treatment of systemic lupus erythematosus (SLE) [1,2,3,4,5,6,7,8]. This allowed the inclusion of belimumab in the 2019 updated European League Against Rheumatism (EULAR) recommendations for SLE management as an approved biological drug to be used in patients refractory to standard of care, which includes glucocorticoids and hydroxychloroquine, with or without concomitant or previous immunosuppressive therapy [9].

The BLISS-LN study [10] provided additional evidence of the efficacy of belimumab in patients with lupus nephritis, which was also confirmed in real-world studies [11].

BeRLiSS (Belimumab in Real Life Setting Study) is the largest European nationwide cohort aimed at investigating belimumab effects on disease activity, damage progression, attainment of remission, and low disease activity (LDA) and assessing predictors of treatment response in SLE patients across Italian references lupus Centers for the treatment of SLE [4,5,11,12,13].

Data from the BeRLiSS cohort showed that a consistent proportion of patients could experience clinical improvement and achieve LDA and remission [12]. In addition, BeRLiSS provided evidence that patients in the early phase of the disease, who initiated belimumab before accruing damage, have the highest chance of having a prompt clinical benefit. On the other hand, BeRLiSS showed that a proportion of patients not achieving a response to belimumab after 6 months of treatment could still attain a response later [12], suggesting that a longer duration of treatment should be granted before considering the drug as ineffective [14].

A pooled analysis from randomized controlled trials BLISS-52 and BLISS-76 [15,16] suggested that belimumab is effective in patients with musculoskeletal and skin involvement [17]. However, firm evidence in clinical practice and quantification through organ-specific indexes is still lacking.

Therefore, we considered all patients with joint and skin involvement enrolled in the BeRLiSS cohort (BeRLiSS-JS) and performed a sub-analysis intending to investigate the efficacy of belimumab in these patients.

Moreover, we also analyzed predictors of long-term response to belimumab in patients with early partial response.

## 2. Patients and Methods

### 2.1. Inclusion Criteria

Data from patients enrolled in the BeRLiSS cohort were retrospectively analyzed. BeRLiSS cohort has already been outlined elsewhere [12].

Patients with joint or skin involvement requiring therapy with belimumab, according to physician judgment with available Disease Activity Score 28 (DAS28) and/or Cutaneous LE Area and Severity Index (CLASI) score [18], were enrolled in this study. Baseline active joint involvement was defined according to SLEDAI-2K (Systemic Lupus Erythematosus Disease Activity Index 2000) specific item (arthritis) and skin involvement according to CLASI ≥ 1.

The study was approved by the University of Padova Ethics Committee (3806/AO/16) and carried out according to Helsinki Declaration. Informed consent regarding personal data treatment was obtained from patients.

### 2.2. Data Collection and Management

Patients were prospectively followed up according to EULAR recommendations [9,19], and participating in this study did not interfere with the daily clinical practice. Anonymized patient data were collected in an ad hoc database since belimumab initiation and regularly updated. DAS28, CLASI activity score, and daily prednisone intake were evaluated at baseline, at 6 and 12 months, and every 12 months thereafter.

All collected data were systematically and regularly evaluated. In case of inconsistencies or missing information, centers were required to amend the data. Patients not fulfilling inclusion criteria and data outside qualitative control were excluded.

### 2.3. Outcome Measures

Set outcomes encompassed achievement of SLEDAI responder index (SRI)-4, clinical remission defined as c-SLEDAI = 0 and prednisone (PDN) ≤ 5 mg/day [20], and LDA defined as SLEDAI ≤ 4, prednisone (PDN) ≤ 7.5 mg/day, and no major organ involvement [21]. Joint response was quantified by DAS28: DAS28 < 2.6 (remission), DAS ≥ 2.6 and <3.2 (LDA); DAS28-20, DAS28-50, and DAS28-70 defined as a decrease of ≥20, ≥50%, and ≥70% from baseline values, respectively. Cutaneous response was quantified through the CLASI, and four main outcomes were considered: CLASI = 0 (remission of skin manifestation), CLASI = 1 (LDA), CLASI-20, CLASI-50, and CLASI-70, defined as a decrease of ≥20, ≥50%, and ≥70% from baseline values, respectively. Achievement of all these outcomes at 6, 12, 24, and 36 months was recorded. Early response was set at 6 months, and late response at 12 and 24 months. Patients not achieving CLASI = 0 or DAS28 remission at month 6 were then considered in the analysis of predictors of late response to belimumab, with CLASI = 0 and DAS28 remission as outcomes.

Patients who discontinued belimumab throughout the follow-up due to inefficacy were counted among the non-responders at the subsequent timepoint after discontinuation. This was done to avoid a bias related to the selection of patients responding to belimumab.

### 2.4. Statistical Analysis

Parametric and non-parametric tests were used according to the types of variables. Comparisons of continuous data with parametric distribution were performed using *t*-test, *t*-test for paired data, and one-way analysis of covariance (ANCOVA) with Bonferroni’s post hoc analysis. Continuous data with non-parametric distribution were analyzed using Wilcoxon’s rank sum test and Wilcoxon’s test for paired data. Comparisons of categorical data were performed using χ^2^ test (Pearson test if indicated). *p*-values less than 0.05 were considered significant.

The following variables collected at baseline (belimumab initiation) and at 6 months were included in the univariate analyses: disease activity pattern (chronic active vs. relapsing-remitting pattern) [22]; concomitant immunosuppressive therapy (yes/no) and/or antimalarials (yes/no), and/or GC therapy (yes/no); prednisone-equivalent dose (categorized into ≤5 mg/day, >5 mg/day, and ≥7.5 mg/day); SLEDAI-2K score, anti-dsDNA antibodies, low complement levels (C3 and C4), smoking status, SDI (categorized into >0, >1, >2); CLASI activity score (categorized as CLASI-20, CLASI-50, CLASI-70, CLASI = 1, and CLASI = 0), and CLASI damage score in patients with skin involvement; DAS28 (categorized as DAS28-20, DAS28-50, DAS28-70, DAS28 remission, DAS28 LDA) in patients with joint involvement. Variables with a *p*-value < 0.2 at univariate analysis were included in the multivariate models.

Two multivariate analyses were performed: the analysis of baseline predictors of response to belimumab at different timepoints (6–12–24 months) and the analysis of predictors of late response to belimumab in patients not achieving DAS28 or CLASI remission at month 6. In the latter analysis, data at 6 months were analyzed as possible predictors of response at 12 and 24 months.

Backward stepwise logistic regression was employed to identify predictors of response at 6, 12, and 24 months, with CLASI = 0, CLASI = 1, DAS28 remission, and DAS28 LDA as dichotomous dependent variables, with significance set at 5%.

Statistical analyses were performed using the SPSS (version 28.0) software (Chicago, IL, USA).

## 3. Results

### 3.1. BeRLiSS Cohorts for Joint and Skin Involvement

Demographic, clinical, and serological variables and concomitant treatment at baseline are reported in Table 1.

### 3.2. Follow-Up Data

A significant decrease in DAS28 and CLASI activity scores, as well as in prednisone intake, was observed during follow-up (Figure 1).

### 3.3. Joint Involvement

According to the inclusion criteria, 328 patients with joint involvement were considered: 15 patients discontinued the drug before achieving a 6-month follow-up (5 cases due to inefficacy, 7 to adverse events, and 3 to loss of follow-up); 11 patients had incomplete data which prevented further analysis, and 22 patients did not complete 6 months at the time of data extraction. Thus, 277 patients were included in the final cohort to evaluate the drug efficacy at 6 months. The number of patients with joint involvement achieving the different timepoints and the number of patients discontinuing the drug due to inefficacy in the 6 months before achieving the timepoint is summarized in Table 2. The mean follow-up period of patients with joint involvement was 23.7 ± 14.3 months.

At baseline (belimumab initiation), 11 (4%) patients with joint involvement had DAS28 > 5.1, 180 (65%) DAS28 ≤ 5.1 and ≥3.2, 53 (19%) DAS28 LDA, and 33 (12%) DAS28 < 2.6. The latter group includes patients with active joint involvement not captured by the DAS28 cut-off for remission.

### 3.4. Efficacy of Belimumab in Patients with Joint Involvement

Among patients with joint involvement, SRI-4 response was achieved by 143 (51.6%), 147 (58.5%), 86 (62.3%), and 46 (64.8%) patients at 6, 12, 24, and 36 months, respectively. In Zen et al. [20], remission was achieved by 62 (22.3%), 84 (33.4%), 41 (29.7%), and 25 (35.2%) patients at 6, 12, 24, and 36 months, respectively. LDA was observed in 103 (37.1%), 121 (48.2%), 77 (55.8%), and 61 (60.5%) patients at 6, 12, 24, and 36 months, respectively. Notably, 57.4% of patients were on PDN ≤ 5 mg/day, and 8.5% were PDN-free at 6 months; these proportions increased to 72.7% and 16.3% at 12 months, to 85.1% and 28.9% at 24 months, and 87.7% and 38.5% at 36 months, respectively.

The proportion of patients achieving DAS28-20, DAS28-50, DAS28-70, DAS28 remission, and DAS28 LDA at different timepoints is reported in Figure 2.

Among the 243 patients with DAS28 ≥ 2.6 at baseline, 109 patients (44.8%) achieved DAS28 remission at 6 months, 116 (50%) at 12 months, 81 (61.4%) at 24 months, and 45 (64.3%) at 36 months. In addition, patients with Boolean remission (meaning Swollen joint 0, Tender joint 0, VAS 1/10, PCR ≤ 1 mg/L) were 12 (4.9%), 26 (11.2%), 24 (18.2%), and 23 (32.8%) at 6, 12, 24, and 36 months, respectively.

### 3.5. Skin Involvement

According to the inclusion criteria, 172 patients with skin manifestations were considered: 9 discontinued the drug before achieving a 6-month follow-up (4 due to inefficacy and 5 due to adverse events), and 16 did not complete 6 months of follow-up at the time of data extraction. Thus, the cohort included 151 patients at baseline. The number of patients with skin involvement achieving the different timepoints and those discontinuing the drug due to inefficacy in the six months before achieving the timepoint is summarized in Table 2. The mean follow-up period in patients with skin involvement was 25.9 ± 15.7 months.

Among patients with skin manifestations at baseline, 17 patients (11.2%) had CLASI > 10, 59 (38.8%) CLASI ≤ 10 and >5, 68 (45,4%) CLASI ≤ 5 and >1, and 7 (4.6%) had CLASI = 1.

### 3.6. Efficacy of Belimumab in Patients with Skin Involvement

Among patients with skin manifestations, remission was achieved by 25 (16.5%), 36 (26.1%), 27 (33.7%), and 18 (36.7%) patients at 6, 12, 24, and 36 months, respectively; LDA was observed in 49 (32.4%), 57 (41.3%), 45 (56.2%), and 34 (69%) patients at 6, 12, 24, and 36 months, respectively. Fifty-four percent of patients were on PDN ≤ 5 mg/day, and 7.8% were PDN-free at 6 months. These percentages increased to 66.1% and 12.1% at 12 months, 81.7% and 19.4% at 24 months, and 85.7% and 26.5% at 36 months, respectively.

The proportion of patients achieving CLASI = 0, CLASI-20, CLASI-50, and CLASI-70 during follow-up is reported in Figure 3.

Patients achieving CLASI = 1 decreased over time due to the progressive achievement of remission (13.6%, 10.2%, 4.3%, and 0% at 6, 12, 24, and 36 months, respectively). The proportion of patients with CLASI > 10 significantly decreased from baseline (14.5%) to 4.8% at 6 months, 0% at 12 months, 1.4% at 24 months, and 0% at 36 months.

Patients with CLASI > 10 at baseline achieved CLASI remission in 1 (4.5%), 4 (19%), and 2 (20%) cases at 6, 12, and 24 months, respectively. CLASI-20 was achieved by 13 (59.1%), 4 (66.7%), and 8 (80%) patients at 6, 12, and 24 months, CLASI-50 by 8 (36.4%), 11 (52.4%), and 6 (60%) patients at 6, 12, and 24 months, and CLASI-70 by 5 (22.7%), 7 (33.3%), and 3 (30%) patients at 6, 12, and 24 months, respectively. This means that a great proportion of patients with high disease activity (CLASI > 10) not achieving remission still experienced a clinically significant improvement in their skin involvement during the follow-up.

### 3.7. Multivariate Analyses of Baseline Predictors of DAS28 Remission, DAS28 LDA and CLASI = 0 at 6, 12, and 24 Months

At multivariate analysis, the predictor of DAS28 remission at 6 months was a lower DAS28 at baseline, and predictors of DAS28 remission at 12 and 24 months were a shorter disease duration and a lower DAS28 at baseline. Similar findings were obtained when DAS28 LDA was used as the outcome measure (Table 3).

A lower CLASI activity at baseline predicted CLASI remission at different timepoints (Table 4).

### 3.8. Patients with Partial Response to Belimumab at 6 Months

When considering patients who did not achieve DAS28 remission at 6 months, 37 out of 142 (26%) and 35 out of 84 (41.6%) achieved remission at 12 and 24 months, respectively. DAS28 LDA at month 6 was associated with remission in these patients at 12 months. Among patients in DAS28 LDA at 6 months, 59.4% achieved remission at 12 months compared to 24.3% of patients not in DAS28 LDA (*p* = 0.034). This result was confirmed in the multivariate model after correction for other variables, including disease duration, serological status, and PDN dose intake (Table 5).

Nevertheless, patients who were not in DAS28-LDA at 6 months could achieve remission later during the follow-up, as 20 out of 39 patients who were not in DAS28-LDA (51.3%) at 6 months were in DAS28 remission at 24 months. Furthermore, no other clinical and serological variables at 6 months were associated with the achievement of DAS28 remission in the long term.

Among patients who did not achieve CLASI = 0 at 6 months, a significant proportion could achieve complete CLASI remission thereafter: 24 out of 92 (26.1%) and 25 out of 54 (46.3%) at 12 and 24 months, respectively. Patients achieving CLASI-50 at 6 months were more likely to achieve CLASI = 0 at 12 (*p* = 0.028) and 24 months (*p* = 0.05). Conversely, CLASI-20 at 6 months was not associated with an increased chance of achieving CLASI = 0 in the following months. However, few patients with CLASI-20 at 6 months could become CLASI = 0 responders during the follow-up (1 out of 5 on average).

At multivariate analysis, a lower CLASI activity score at 6 months was the only predictor of CLASI = 0 at 12 months (Table 6).

## 4. Discussion

In this study, we confirmed in clinical practice data from randomized control trials showing the high effectiveness of belimumab in patients with joint and skin involvement [17].

Belimumab is a human immunoglobulin G1k monoclonal antibody that binds the soluble BLyS, thus preventing the binding to its receptors on B cells. As a result, belimumab inhibits the survival of B cells, including autoreactive B cells, and reduces the differentiation of B cells into Ig-producing plasma cells [23]. In SLE, systemic over-expression of BLyS was reported in some studies [24,25]. In addition, local over-expression of BlyS and its receptor 3 (BR3; also known as BAFF-R) was demonstrated in different tissues, including lesional keratinocytes, kidney-derived cells, and infiltrating B cells in skin and kidney biopsies from lupus patients [26,27]. All this evidence supports the rationale for the use of belimumab in SLE.

In keeping with these observations, in our multicenter cohort, a third of refractory SLE patients achieved remission according to different disease activity indices, including SLEDAI-2K, DAS28, and CLASI. In addition, almost half of the patients could achieve LDA after 12 months of belimumab therapy.

We also showed the glucocorticoid-sparing effect of belimumab. We observed a decrease in daily PDN dose intake and a consistent proportion of patients who could withdraw PDN during the follow-up. Interestingly, the proportion of glucocorticoid-free patients increased during the follow-up in both cohorts of patients with joint and skin involvement and was observed in about one-third of patients at 36 months (Figure 4). Notably, the decrease in the cumulative intake of glucocorticoids is one of the main goals in the modern management of SLE [2] in order to prevent long-term damage accrual [20,28,29,30] and quality of life [31,32] not only in patients with joint and skin manifestations but also in patients with severe features of SLE [33].

In our multivariate models, a short disease duration and a lower DAS28 or CLASI at baseline predicted DAS28 and CLASI remission or LDA during the follow-up, suggesting that the earlier the use of belimumab, the better the outcome, regardless of the organ-specific activity score used to quantify organ-specific activity. In fact, this is in line with our previous studies, which demonstrated that the use of belimumab in the early stage of the disease was associated with a better SRI-4 response and damage prevention [5,11,12].

Another critical aspect in the evaluation of drug effectiveness is the choice of outcome measures. A too-stringent response criteria might have contributed to several RCT failures [34,35,36,37]. In this paper, we considered not only organ-specific outcome measures but also the improvement in 20%, 50%, and 70% of DAS28 and CLASI as potential clinical targets. However, DAS28-20 and DAS28-70 do not seem appropriate outcomes being DAS28-20 too liberal and DAS28-70 too stringent. In addition, it has to be pointed out that patients with a low DAS28 at baseline can achieve remission without fulfilling 50% or 70% improvement in DAS28. Finally, it is worth noting that a non-negligible number of patients with active joint involvement had a DAS28 < 2.6 at baseline, meaning that DAS28 cannot always capture lupus joint manifestations, which can occur in ankles and feet or with tendinitis or bursitis.

On the other hand, in patients with skin involvement and high disease activity (CLASI > 10), where the achievement of remission is more difficult, CLASI-50 could be a reasonable and clinically meaningful outcome, especially in the short term [38].

An open question in the belimumab treatment is when a patient can be considered a “non-responder” and, in turn, when belimumab should be discontinued [2]. We found that the response or a significant improvement in DAS28 or CLASI at 6 months predicted the response at 12 months. Still, even 24.3% of patients with joint and 26.1% of patients with skin manifestations who were non-responders at 6 months achieved full response at 12 months and 46.3% and 51.3% at 24 months, respectively. In patients with skin involvement, the lack of CLASI remission early during belimumab treatment did not prevent its achievement later during the follow-up, suggesting that 6 months may not be enough to fully evaluate belimumab efficacy [5,12]. According to our results, patients with mild improvement, such as those with CLASI-20 at 6 months, could benefit from belimumab treatment later during the follow-up. However, patients showing an earlier and higher improvement have a greater probability of achieving remission. Thus, our data suggest that one year might be the adequate time span of belimumab therapy needed to thoughtfully evaluate its efficacy. This observation is in keeping with other real-life clinical studies [12] and clinical trials [6,39,40]. Furthermore, it supports the rationale of continuing belimumab, even in patients who have not experienced a full response at 6 months.

Our study has both strengths and limitations. Limitations include the lack of a control group population, the exclusion of patients for whom data were unavailable at any given timepoints, and the absence of Patient-Reported Outcomes (PROs). These limitations are mainly connected to the retrospective nature of the study, which poses some objective restrictions to the amount of data that can be inferred. The greatest strengths of the study are the real-life setting, the large cohort of patients analyzed, and the long follow-up duration.

## 5. Conclusions

In conclusion, this study confirms the effectiveness of belimumab in SLE patients with joint and skin involvement and reinforces the previous findings suggesting that the early use of belimumab can maximize its efficacy.

A glucocorticoid-sparing effect of belimumab was also shown, with a decrease in daily dose and a consistent proportion of patients who become PDN-free during the follow-up, lowering the cumulative glucocorticoid intake.

Finally, patients with early partial improvement have a significant chance of achieving organ-specific remission later during the follow-up supporting the rationale for continuing belimumab, even in patients who have not experienced a full response at 6 months.

## Figures and Tables

**Figure 1 jpm-13-00691-f001:**
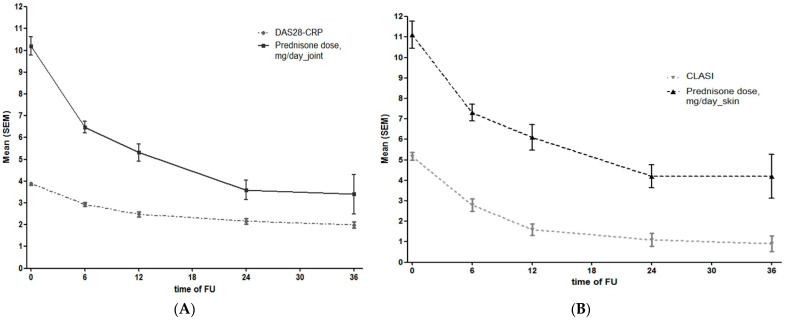
(**A**) DAS28 and glucocorticoid intake during the follow-up in patients with joint involvement at different timepoints. (**B**) CLASI and glucocorticoid intake during the follow-up in patients with skin involvement at different timepoints.

**Figure 2 jpm-13-00691-f002:**
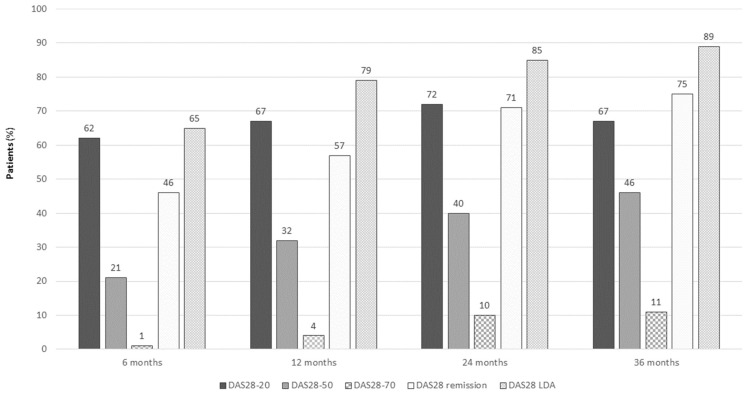
DAS28-20, DAS28-50, DAS28-70, DAS28 remission, and DAS20 LDA in SLE patients on belimumab at different timepoints.

**Figure 3 jpm-13-00691-f003:**
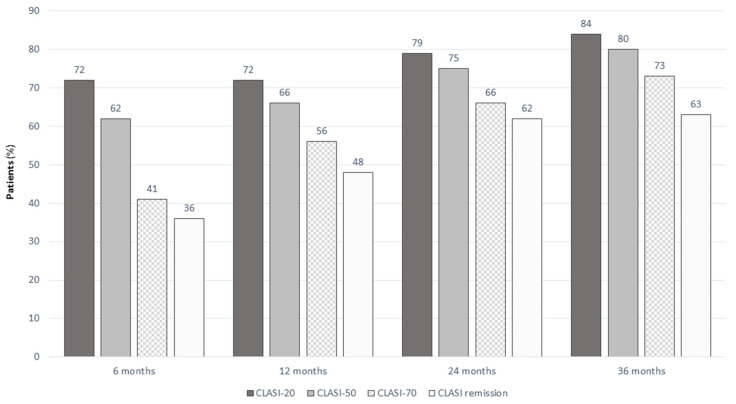
CLASI-20, CLASI-50, CLASI-70, and CLASI remission in SLE patients treated with belimumab at different timepoints.

**Figure 4 jpm-13-00691-f004:**
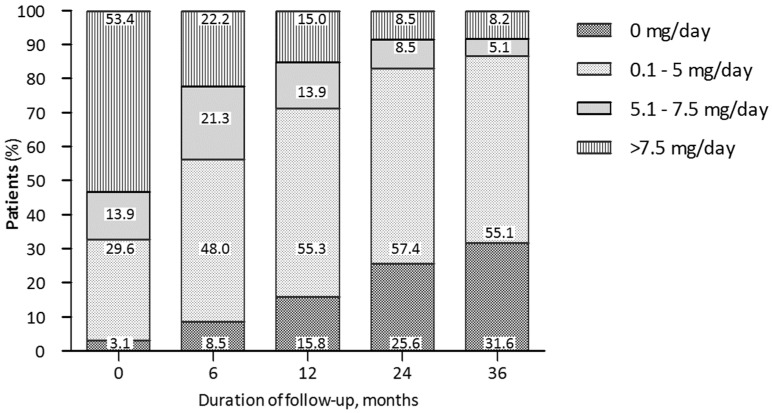
Trends in daily dose PDN intake during belimumab treatment in patients with joint and skin manifestations at different timepoints.

**Table 1 jpm-13-00691-t001:** Baseline demographic, clinical, and serological variables in SLE patients with joint and skin involvement at baseline treated with belimumab.

	Joint Involvement	Skin Involvement
Total patients, n (%)	277 (100)	151 (100)
Female, n (%)	255 (92)	140 (92.7)
Age at diagnosis, mean ± SD years	30.22 ± 11.6	29.3 ± 11.1
Age at belimumab initiation, mean ± SD years	42.22 ± 11.8	40.6 ± 10.4
Disease duration, mean ± SD years	12 ± 9	11 ± 9
Disease duration ≤ 2 years, n (%)	47 (17)	28 (18.5)
Total months follow-up, mean ± SD	23.42 ± 14.44	25.2 ± 15.9
SLEDAI-2K score, mean ± SD	10 ± 3.0	10.0 ± 4.0
SLEDAI-2K ≥ 10, n (%)	129 (46.6)	93 (61.6)
c-SLEDAI, mean ± SD	6.23 ± 2.8	6.58 ± 3.4
SDI, median (25–75%)	1 (0–2)	1 (0–1)
Chronic active pattern, n (%)	100 (36.1)	61 (40.4)
Smoking, n (%)	51 (19.6)	32 (21.9)
Antiphospholipid syndrome, n (%)	44 (16.2)	16 (10.9)
CLASI, median (IQR)	-	4 (3–7)
DAS28, mean ± SD	3.9 ± 4.2	-
**Serology**		
Anti-dsDNA, n (%)	222 (90.7)	125 (82.8)
Anti-Sm, n (%)	70 (25.4)	45 (30)
Anti-SSA, n (%)	125 (45.3)	84 (56)
Anti-SSB, n (%)	49 (17.8)	38 (25.3)
Anti-U1RNP, n (%)	87 (31.5)	53 (35.3)
Anti-P-ribosomal	17(6.2)	11 (7.3)
Antiphospholipid, n (%)	95 (34.7)	43 (28.7)
Low C3, mean ± SD (range)	72.05 ± 22.83	70.01 ± 25.50
Low C4, mean ± SD (range)	11.25 ± 6.38	11.54 ± 7.67
**Concomitant treatment**		
Oral glucocorticoids		
Daily PDN intake, mean ± SD mg	10.15 ± 6.92	11.05 ± 8.15
Daily PDN intake ≤ 5 mg, n (%)	92 (33.2)	41 (27.2)
Antimalarials, n (%)	184 (66.4)	110 (72.8)
Immunosuppressants, n (%)	178 (64.3)	97 (64.2)

SLE: systemic lupus erythematosus; SLEDAI-2K: SLE Disease Activity Index 2000; c-SLEDAI: Clinical SLEDAI-2k; dsDNA: double-stranded DNA; PDN: prednisone equivalent; SD: standard deviation; SDI: SLICC Damage Index; CLASI: Cutaneous LE Area and Severity Index; DAS28: Disease activity score 28-CRP.

**Table 2 jpm-13-00691-t002:** Number of SLE patients with joint or skin involvement considered in the analyses at different timepoints, including those in follow-up and those who discontinued the drug due to inefficacy in the 6 months before achieving the timepoint.

	Joint Involvement		Skin Involvement	
Months	Patients in Follow-Up *	Patients Discontinuing Belimumab in the 6 Months before the Timepoint **	Patients in Follow-Up *	Patients Discontinuing Belimumab in the 6 Months before the Timepoint **
6 months	272	5	147	4
12 months	215	36	118	20
24 months	114	24	69	11
36 months	59	15	42	7
48 months	28	4	23	3

* Number of patients achieving the 6, 12, 24, 36, and 48 months of follow-up. ** Due to inefficacy.

**Table 3 jpm-13-00691-t003:** Baseline predictors of DAS28 remission (A) and LDA (B) at 6, 12, and 24 months (multivariate analysis).

**A**						
	**DAS28 Remission at 6 Months**		**DAS28 Remission at 12 Months**		**DAS28 Remission at 24 Months**	
	**OR (95% CI)**	***p*-Value**	**OR (95% CI)**	***p*-Value**	**OR (95% CI)**	***p*-Value**
**DAS28**	1.02 (0.38–2.74)	0.960	** 0.56 (0.427–0.735) **	** <0.001 **	** 0.54 (0.37–0.79) **	**0.001**
**Disease Duration**	1.09 (0.96–1.23)	0.168	** 0.94 (0.91–0.99) **	** 0.008 **	1.030 (0.97–1.09)	0.330
**Low C3**	0.98 (0.93–1.04)	0.551	1.002 (0.99–1.02)	0.733	0.99 (0.98–1.02)	0.812
**Smoke**	6.90 (0.63–72.86	0.113	0.52 (0.23–1.17)	0.115	-	-
**SLEDAI-2K**	1.173 (0.816–1.686)	0.389	0.92 (0.83–1.02)	0.128	0.88 (0.74–1.04)	0.135
**B**						
	**DAS28 LDA at 6 Months**		**DAS28 LDA at 12 Months**		**DAS28 LDA at 24 Months**	
	**OR (95% CI)**	***p*-Value**	**OR (95% CI)**	***p*-Value**	**OR (95% CI)**	***p*-Value**
**DAS28**	**1.04 (0.91–1.19)**	**<0.001**	**0.59 (0.44–0.80)**	**<0.001**	**0.57 (0.39–0.89)**	**0.011**
**Disease Duration**	0.99 (0.95–1.04)	0.751	**0.94 (0.91–0.99)**	** 0.008 **	0.99 (0.92–1.07)	0.783
**Low C3**	1.01 (0.99–1.03)	0.244	1.00 (0.99–1.02)	0.576	0.99 (0.97–1.01)	0.458
**Smoke**	**2.69 (1.03–7.023)**	**0.044**	0.87 (0.34–2.21)	0.076	-	-
**SLEDAI-2K**	1.04 (0.91–1.19)	0.536	0.95 (0.91–0.99)	0.412	1.05 (0.85–1.29)	0.669

DAS28: disease activity score 28 CRP; C3: complement fraction C3; SLEDAI-2K: SLE Disease Activity Index 2000.

**Table 4 jpm-13-00691-t004:** Baseline predictors of CLASI = 0 at 6, 12, and 24 months (multivariate analysis).

	CLASI Remission at 6 Months		CLASI Remission at 12 Months		CLASI Remission24 Months	
	OR (95% CI)	*p*-Value	OR (95% CI	*p*-Value	OR (95% CI)	*p*-Value
**CLASI**	**0.71 (0.60–0.83)**	**<0.001**	** 0.83 (0.73–0.93) **	**0.002**	0.78 (0.64–0.95)	0.014
**Disease Duration**	**0.94 (0.89–0.93)**	** 0.026 **	0.98 (0.93–1.03)	0.397	0.91 (0.87–1.00)	0.057
**SLEDAI 2K**	1.08 (0.97–1.21)	0.244	1.01 (0.89–1.15)	0.875	0.99 (0.81–1.22)	0.997
**Anti-dsDNA**	-	-	1.82 (0.58–5.73)	0.308	0.73 (0.14–3.86)	0.715
**Low C3**	-	-	**3.11 (1.10–8.92)**	**0.033**	2.13 (0.46–9.80)	0.331
**HCQ baseline**	0.70 (0.29–1.68)	0.424	0.98 (0.38–2.54)	0.969	0.92 (0.23–3.77)	0.912

CLASI: Cutaneous LE Area and Severity Index; SLEDAI-2K: SLE Disease Activity Index-2000; HCQ: hydroxycholoroquine.

**Table 5 jpm-13-00691-t005:** Six-month predictors of long-term DAS28 remission in patients with partial response to belimumab.

Variables at 6 Months	DAS28 Remission at 12 Months		DAS28 Remission at 24 Months	
	OR (95% CI)	*p*-Value	OR (95% CI)	*p*-Value
**DAS28 < 3.2**	**2.69 (1.03–7.00)**	**0.043**	1.07 (0.99–1.16)	0.067
**Disease Duration**	0.97 (0.94–1.02)	0.269	**0.94 (0.91–0.99)**	** 0.013 **
**Low C3**	0.99 (0.97–1.01)	0.222	0.98 (0.950–1.01)	0.163
**Low C4**	0.97 (0.91–1.03)	0.348	1.05(0.96–1.16)	0.291
**SLEDAI-2K**	0.95 (0.81–1.11)	0.497	0.83 (0.67–1.03)	0.089
**Anti-dsDNA**	1.51 (0.45–4.99)	0.502	2.88 (0.43–1.92)	0.275
**Prednisone**	0.94 (0.84–1.06)	0.330	0.92 (0.76–1.11)	0.380

DAS28: disease activity score 28 CRP; SLEDAI-2K: SLE Disease Activity Index 2000; C3/C4: complement fraction C3/C4; dsDNA: double-stranded DNA.

**Table 6 jpm-13-00691-t006:** Six-month predictors of long-term cutaneous remission in patients with partial response to belimumab.

Variables at 6 Months	CLASI Remission at 12 Months		CLASI Remission at 24 Months	
	OR (95% CI)	*p*-Value	OR (95% CI)	*p*-Value
**CLASI**	0.69 (0.49–0.98)	0.041	0.74 (0.49–1.11)	0.145
**Low C3**	0.99 (0.97–1.02)	0.658	0.98 (0.94–1.02)	0.576
**Low C4**	0.94 (0.86–1.03)	0.199	0.96 (0.87–1.07)	0.481
**Anti-dsDNA**	2.33 (0.48–11.28)	0.292	3.80 (0.41–34.87)	0.238

CLASI: Cutaneous LE Area and Severity Index; C3/C4: complement fraction C3/C4; dsDNA: double-stranded DNA. Finally, no baseline demographic, clinical, or serological features could predict early (6 months) vs. late (12 months) remission, both in patients with joint and skin involvement.

## Data Availability

Data are available upon reasonable request to the corresponding author.

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
