# Peer review of "Early and Late Response and Glucocorticoid-Sparing Effect of Belimumab in Patients with Systemic Lupus Erythematosus with Joint and Skin Manifestations: Results from the Belimumab in Real Life Setting Study—Joint and Skin (BeRLiSS-JS)"

_jpm, 2023, doi:10.3390/jpm13040691_

Round 1

Reviewer 1 Report

Comments and Suggestions for Authors

The manuscript by Zen et al. investigated the effectiveness of belimumab in SLE patients with joints and skin involvement enrolled in the BeRLiSS trial. The statistical analysis with various clinical outcomes confirms the effectiveness of belimumab in SLE patients specifically with the joints and skin involvement. The current study results are in line with previous findings where belimumab treatment in SLE patients has shown improvement with organ damage as in Lupus Nephritis patients. Additionally, the study reinforces the early use of belimumab in maximizing efficacy and favorable outcomes in SLE patients.

Major suggestions on the manuscript:

The paper does not provide any explanation on mechanistic aspects of how belimumab could have contributed to improvement of joints and skin manifestations in SLE patients. The authors can speculate based on literature and discuss these point in the discussion section.

As belimumab is directed against the cytokine BAFF, assessing plasma levels of BAFF and performing correlations with various clinical outcomes mentioned in the study can yield useful outcomes in this well-designed study with large cohort of SLE patients.

Minor suggestions on the manuscript:

All figures quality needs to be improved. The figures are blurry.

Maintain the fonts of X and Y axis same for all figures. The fonts in Figure 4 are different from other figures.

Manuscript is well written

Reviewer 2 Report

In this study, the authors confirmed in clinical practice data from randomized control trials showing a high effectiveness of belimumab in lupus patients with joint and skin involvement.

This is a very well-designed study, with excellent statistical analysis, appropriate figures and tables, and harmonious and coherent writing. Indeed, this study confirms the effectiveness of belimumab in systemic lupus erythematosus patients with joint and skin involvement and reinforces the previous findings suggesting that the early use of belimumab can maximize its efficacy.
